# Dimethyl Sulfoxide as a Biocompatible Extractant for Enzymatic Bioluminescent Toxicity Assays: Experimental Validation and Molecular Dynamics Insights

**DOI:** 10.3390/toxics13121038

**Published:** 2025-11-30

**Authors:** Oleg S. Sutormin, Victoria I. Lonshakova-Mukina, Anna A. Deeva, Alena A. Gromova, Ruslan Ya. Bajbulatov, Valentina A. Kratasyuk

**Affiliations:** 1Scientific and Educational Center, Institute of Nature and Technical Sciences, Surgut State University, 628412 Surgut, Russia; vlonshakova@sfu-kras.ru (V.I.L.-M.); adeeva@sfu-kras.ru (A.A.D.); al.groma@yandex.ru (A.A.G.); bajbulatov_rya@surgu.ru (R.Y.B.); vkratasyuk@sfu-kras.ru (V.A.K.); 2Department of Biophysics, School of Fundamental Biology and Biotechnology, Siberian Federal University, 660041 Krasnoyarsk, Russia; 3Photobiology Laboratory, Federal Research Center ‘Krasnoyarsk Science Center SB RAS’, Institute of Biophysics, Russian Academy of Sciences, Siberian Branch, 50/50 Akagemgorodok, 660036 Krasnoyarsk, Russia

**Keywords:** dimethyl sulfoxide (DMSO), bioluminescent enzymatic assay, diesel-contaminated soils, ecotoxicological assessment, molecular dynamics simulation, NAD(P)H:FMN-oxidoreductase–luciferase system

## Abstract

Diesel fuel is among the most persistent petroleum-derived pollutants in soils, posing long-term ecological and toxicological risks, especially in cold-climate regions where natural degradation is limited. Reliable assessment of diesel-contaminated soils remains difficult because conventional solvent-based analyses are incompatible with bioassays, while aqueous extracts underestimate hydrocarbon toxicity. This study evaluated dimethyl sulfoxide (DMSO) as a biocompatible extractant for enzymatic bioluminescent toxicity assays employing the coupled NAD(P)H:FMN-oxidoreductase and bacterial luciferase (BLuc–Red) system. Soil samples artificially contaminated with diesel fuel were analyzed using DMSO extracts in combination with molecular dynamics (MD) simulations to examine enzyme stability in solvent environments. Moderate DMSO concentrations (4–6% *v*/*v*) maintained enzymatic activity, whereas higher levels caused partial inhibition. Diesel hydrocarbons dissolved in DMSO strongly suppressed luminescence, and soil extracts exhibited a clear dose–response relationship between contamination level and enzymatic inhibition. MD simulations confirmed that neither DMSO nor diesel induced large-scale unfolding of luciferase or reductase, though localized flexibility changes and partial dehydration of active site residues was observed, which may account for the detected inhibition of luminescence at higher DMSO concentrations. These results demonstrate that DMSO provides an effective and biocompatible extraction medium for enzymatic bioluminescent assays, enabling accurate toxicity evaluation of petroleum-contaminated soils and offering a promising tool for ecotoxicological risk assessment in oil-impacted environments.

## 1. Introduction

Diesel fuel, as a refined petroleum product, represents one of the most widespread sources of soil contamination in industrial and oil-producing regions. Along with crude oil and other petroleum hydrocarbons, diesel residues persist in terrestrial ecosystems, altering soil physicochemical properties, suppressing microbial activity, and impairing plant growth [1,2,3,4]. These effects result in long-term ecological and toxicological risks, particularly in cold-climate regions where natural degradation processes are markedly slow [5]. In Western Siberia, including the Khanty-Mansi Autonomous Okrug-Yugra—one of the world’s largest centers of oil production—frequent spills and leakages are exacerbated by harsh environmental conditions, making soil remediation highly challenging [4,6]. Under such circumstances, developing reliable and sensitive tools for assessing the ecological risks of diesel-contaminated soils is an urgent priority.

Conventional chemical analyses of petroleum hydrocarbons are typically based on solvent extraction followed by chromatographic or gravimetric determination [7,8]. While these approaches provide quantitative estimates of contaminant levels, they rely on aggressive solvents such as hexane, chloroform, or carbon tetrachloride which are toxic, environmentally hazardous, and incompatible with biological systems [9]. Conversely, bioassays often employ water as an extractant, yet diesel hydrocarbons are poorly soluble in aqueous media, leading to an underestimation of their toxicity [10,11]. This discrepancy complicates direct comparison between chemical and biological assessments of contaminated soils, creating a methodological gap in environmental monitoring.

Biotesting strategies, particularly those based on luminescence, offer an integrated measure of toxicity and represent promising alternatives to chemical analysis [12,13]. However, organism-based assays using algae, protozoa, or crustaceans are time-consuming, require continuous culture maintenance, and are prone to biological variability. Bacterial bioluminescent assays are faster but still depend on living cells, which can yield ambiguous responses to toxicants [13,14]. To overcome these limitations, cell-free enzymatic bioluminescent assays have been developed using the coupled NAD(P)H:FMN-oxidoreductase and bacterial luciferase (BLuc–Red) system, which combines sensitivity, reproducibility, and methodological simplicity [14]. This approach eliminates the need for live cultures and has been successfully applied for toxicity monitoring of water, air, snow, and soil [14,15].

A critical methodological challenge in adapting enzymatic assays to soil samples lies in selecting an appropriate extractant. The solvent must effectively solubilize hydrophobic diesel hydrocarbons while maintaining compatibility with enzymatic systems. Dimethyl sulfoxide (DMSO) is a promising candidate due to its unique physicochemical properties. As a polar aprotic solvent, DMSO is widely used in biochemistry and pharmacology to solubilize hydrophobic compounds while preserving enzymatic activity at moderate concentrations [16,17]. Its relatively low toxicity and full miscibility with water suggest that it could serve as a bridge between chemical extraction and bioassay-based toxicity assessment of diesel-contaminated soils [18].

Despite these advantages, systematic evaluation of DMSO in enzymatic bioluminescent assays for diesel-contaminated soils remains limited. Moreover, it is critical to determine whether DMSO itself perturbs the structural stability of the BLuc–Red enzymatic system, as solvent-induced conformational changes could compromise assay reliability. Molecular dynamics (MD) simulations offer a powerful means of addressing this issue by enabling atomic-level examination of protein dynamics in the presence of solvents and hydrocarbon molecules [19,20].

The present study was designed to evaluate the suitability of DMSO as a biocompatible extractant for enzymatic bioluminescent assays aimed at assessing the toxicity of diesel-contaminated soils. We combined experimental assays with MD simulations to investigate whether DMSO and diesel hydrocarbons alter the structural stability of NAD(P)H:FMN-oxidoreductase and bacterial luciferase—the key enzymes of the BLuc–Red system. We hypothesized that DMSO, while enhancing hydrocarbon solubilization, would not induce significant conformational changes in these enzymes, thereby preserving assay validity. This integrative approach provides both methodological validation and mechanistic insight, contributing to the advancement of ecotoxicological monitoring tools for soils impacted by petroleum hydrocarbons.

## 2. Materials and Methods

### 2.1. Soil Sampling and Characterization

Soil material was obtained from a boreal forest site located near the settlement of Nizhnesortymsky (62°45′38″ N, 71°78′86″ E) in the Surgut District of the Khanty-Mansi Autonomous Okrug-Yugra, Western Siberia. To obtain a representative sample of the upper soil horizon, five individual cores were taken within a small area (0–40 cm depth) and combined into a bulk composite. The resulting mixture (approximately 5 kg) was classified as Albic Podzolic soil according to the current Russian soil classification scheme [21].

The collected material displayed characteristics typical of podzolic soils in cold-climate coniferous ecosystems: a sandy-loam texture, low natural acidity, and a limited amount of organic matter. Before analytical procedures, the soil was air-dried under laboratory conditions for two days and gently homogenized by passing it through a 1 mm sieve to remove coarse fragments.

Routine physicochemical parameters were evaluated following nationally accepted analytical protocols [22,23,24]. Organic matter content was determined by dichromate oxidation with titrimetric quantification. Soil acidity (pH) was measured potentiometrically in aqueous suspensions prepared at a 1:5 soil-to-water ratio. Electrical conductivity and the proportion of dissolved solids in the water extract were assessed after drying the extract at 105 °C and measuring conductivity with a calibrated meter.

Total petroleum hydrocarbons (TPH) were quantified using the gravimetric procedure described in standard PND F 16.1:41-04 [25]. Briefly, 30–40 g of soil was repeatedly extracted with chloroform until the solvent became clear. The combined extracts were evaporated, and the remaining hydrocarbon residue was weighed and expressed relative to the dry soil mass.

### 2.2. Preparation of Diesel-Contaminated Samples

Commercial diesel fuel (density 0.818 g/cm^3^) from the Surgut Condensate Stabilization Plant (KhMAO–Yugra, Russia) was used as a model pollutant. Soil samples were artificially contaminated at three concentrations: 5, 10, and 20 g/kg. These levels were selected to represent moderate to high contamination, with 20 g/kg considered moderate relative to the regional maximum permissible concentration of 20 mg/kg [26]. Uniform mixing of diesel fuel with soil was achieved by continuous stirring in a sealed vessel using a ball mill activator (1 revolution per second, 24 h). Subsequent sample preparation followed the gravimetric method for petroleum hydrocarbons in soils (PND F 16.1:41-04) [25].

### 2.3. Bioluminescent Enzymatic Assay

The bioluminescence inhibition assay was employed to determine the integral toxicity of soil samples, based on measuring the activity of the coupled NAD(P)H:FMN-oxidoreductase and bacterial luciferase (BLuc–Red) system.

Reaction mixtures (420 µL total volume) contained 150 µL of 0.05 M potassium phosphate buffer (pH 6.9), 10 µL of enzyme solution, 50 µL of 0.0025% (*v*/*v*) myristic aldehyde (C_14_), 50 µL of 0.4 mM NADH, 150 µL of the test sample, and 10 µL of 0.5 mM FMN. Soil extracts prepared with 25% (*v*/*v*) DMSO were used directly in the reaction mixture without additional dilution, resulting in a final DMSO concentration of 9% in the assay (calculated based on 150 μL extract in the total reaction volume of 420 μL). Enzyme preparations included luciferase (0.5 mg/mL, recombinant *Escherichia coli*) and NAD(P)H:FMN oxidoreductase (0.3 activity units, *Vibrio fischeri*), both produced at the Institute of Biophysics SB RAS (Krasnoyarsk, Russia).

Luminescence intensity was recorded using a GloMax 20/20^n^ luminometer (Promega, Madison, WI, USA). Residual luminescence was expressed as (I/I_0_) × 100%, where I and I_0_ are the luminescence intensities of test and control samples, respectively. Toxicity was classified as follows: >80%—no impact; 50–80%—moderate impact; <50%—significant impact.

### 2.4. Molecular Dynamics Simulations of the BLuc–Red System

Classical molecular dynamics (MD) simulations were performed using GROMACS 2024.4 with the CHARMM36 force field. The structure of *Photobacterium leiognathi* luciferase was prepared from a previously published model [27], while the structure of *Aliivibrio fischeri* NAD(P)H:FMN-oxidoreductase was obtained from the Protein Data Bank (PDB ID: 1VFR). Topology and parameter files for the FMN cofactor were generated using the CHARMM General Force Field (CGenFF 3.0) (SilcsBio LLC, Baltimore, MD, USA, 2023).

For each enzyme, three systems were prepared: (i) protein + water + ions; (ii) protein in aqueous DMSO (10 *w*/*w* %); and (iii) protein in aqueous DMSO (10 *w*/*w* %) with diesel components (1.5 *w*/*w* %). The diesel model included representative hydrocarbons based on Pires de Oliveira et al. [28], consisting of n-paraffins, isoparaffins, naphthenes, aromatics, and heteroaromatics. These components were incorporated as individual molecules at a total mass fraction of 1.5%.

Energy minimization was carried out using the steepest descent algorithm, followed by two equilibration steps under NVT and NPT ensembles (300 K, 1 bar) using the V-rescale thermostat and Parrinello–Rahman barostat. Three independent production runs of 100 ns each were performed for every system. The simulation results were analyzed using built-in GROMACS tools to obtain RMSD, Rg, SASA, and RMSF profiles.

All simulation input files and parameter sets are available from the corresponding author upon reasonable request.

### 2.5. Data Processing and Statistical Analysis

All experiments were performed in triplicate. Data are presented as mean ± standard deviation (SD). Statistical significance was accepted at *p* < 0.05. Computational data were processed using GROMACS built-in tools.

## 3. Results

To evaluate the applicability of DMSO as a solvent for bioluminescent enzymatic assays of petroleum-contaminated soils, a series of experiments was performed combining biochemical testing and molecular dynamics simulations. The results are presented in four parts: (i) the effect of DMSO on the activity of the BLuc–Red system, (ii) the influence of diesel fuel dissolved in DMSO, (iii) the effect of soil extracts obtained with DMSO, and (iv) structural insights from molecular dynamics simulations of luciferase and reductase.

### 3.1. Effect of DMSO Concentration on BLuc–Red Enzymatic Activity

The influence of DMSO on enzymatic activity was evaluated by measuring the maximum luminescence intensity of the BLuc–Red system at increasing solvent concentrations (Figure 1). DMSO at 4–6% (*v*/*v*) produced no measurable inhibition, while higher concentrations caused a progressive, dose-dependent decrease in luminescence intensity. At 9% DMSO, luminescence was reduced approximately 1.6-fold compared to the control. Despite this reduction, residual activity remained sufficient for inhibition-based bioassays, indicating that moderate DMSO concentrations are compatible with the enzymatic system.

### 3.2. Effect of Diesel Fuel Dissolved in DMSO

To assess the combined impact of hydrocarbons and solvent, diesel fuel was dissolved in 10% (*v*/*v*) DMSO and tested using the BLuc–Red system (Figure 2). A clear concentration-dependent inhibition of luminescence was observed. Even at 0.01% diesel fuel, the luminescence intensity decreased approximately 1.8-fold compared with the control. These findings demonstrate that the BLuc–Red system effectively detects petroleum hydrocarbons solubilized in DMSO, confirming its applicability for integral toxicity assessment of hydrophobic contaminants.

### 3.3. Toxicity of Soil Extracts from Contaminated Samples

The inhibitory effects of soil extracts were analyzed for uncontaminated and diesel-spiked samples extracted with 25% DMSO (*v*/*v*). The extracts were used directly in the reaction mixture without additional dilution, yielding a final DMSO concentration of 9% in the assay (calculated based on the addition of 150 μL of extract to a total reaction volume of 420 μL). A strong dose-dependent relationship was observed between contamination level and enzymatic inhibition (Figure 3). For example, soil containing 20 g/kg diesel fuel reduced BLuc–Red activity by approximately 70% relative to the uncontaminated control. This correlation indicates that DMSO-based soil extracts reliably reflect both the degree of contamination and the associated toxicity of petroleum hydrocarbons.

### 3.4. Molecular Dynamics Simulations of Luciferase and Reductase

Molecular dynamics (MD) simulations were conducted to investigate protein structural dynamics under three solvent environments: water, 10 (*w*/*w*) % DMSO aqueous solution, and 10 (*w*/*w*) % DMSO containing 1.5 (*w*/*w*) % diesel fuel.

Three structural parameters—root mean square deviation (RMSD), radius of gyration (Rg), and solvent-accessible surface area (SASA)—were used to evaluate global protein stability (Table 1). No significant changes in RMSD or Rg were detected for either enzyme upon addition of DMSO or diesel fuel. A slight increase in SASA was observed for luciferase in the diesel-containing system (from 297 to 302 × 10^2^ Å^2^), indicating a minor decrease in overall protein compactness.

The RMSF profiles of Cα atoms were then calculated to assess residue flexibility under different conditions. The ΔRMSF = RMSF (DMSO or DMSO + diesel) − RMSF (water) values showed localized rigidity changes (Figure 4). For luciferase, residues 145–155 and 281–287 exhibited reduced mobility in both solvent systems, corresponding to loops involved in dimerization and flavin-binding, respectively. The β-subunit regions remained largely unaffected (Appendix A).

For NAD(P)H:FMN-oxidoreductase, moderate flexibility alterations were observed in residues 90–134 of subunit A and in the C-terminal loop of subunit B (Figure 5a,b). The latter corresponds to a region involved in phosphate fixation of the FMN cofactor, suggesting that solvent composition slightly affects the redox center dynamics.

Analysis of intra- and intermolecular hydrogen bonds revealed a modest decrease in hydrogen bonding between proteins and water upon DMSO addition (Figure 6a,b and Appendix A). This reduction suggests competition of DMSO with water for hydrogen-bonding sites on the protein surface, leading to partial dehydration. In diesel-containing systems, total hydrogen bonds remained lower than in water, but an increase in side-chain–side-chain contacts was observed, indicating slight rearrangement of hydrogen-bond networks under more hydrophobic conditions. Notably, when hydrogen bonds involving the active center were examined separately (Figure 6a,b and Appendix A), a similar trend was observed: the number of hydrogen bonds formed between the active site residues and water decreased in the presence of DMSO in both systems: with diesel compounds and without them. In addition, DMSO molecules were found to penetrate the active center and form several hydrogen bonds with catalytic residues, further contributing to reduced local hydration.

## 4. Discussion

The present study evaluated the applicability of dimethyl sulfoxide (DMSO) as a biocompatible extractant for enzymatic bioluminescent assays aimed at assessing the toxicity of diesel-contaminated soils, integrating experimental data with MD simulations. The results demonstrate that DMSO at moderate concentrations does not significantly inhibit the activity of the coupled NAD(P)H:FMN-oxidoreductase and bacterial luciferase (BLuc–Red) system, while efficiently dissolving hydrophobic diesel hydrocarbons. These findings confirm the initial hypothesis that DMSO can bridge the methodological gap between chemical and biological assessments of hydrocarbon contamination by acting as both an efficient extractant and a biocompatible solvent.

Different DMSO concentrations were intentionally used for distinct methodological tasks. Low concentrations (4–6% *v*/*v*) were applied to assess the intrinsic tolerance of the BLuc–Red system to the solvent. A 10% DMSO fraction was required to dissolve diesel components for controlled evaluation of hydrocarbon-induced inhibition. In contrast, 25% DMSO was selected for soil extraction, as higher solvent strength improves mobilization of hydrophobic contaminants from the soil matrix while still allowing sufficient dilution before enzymatic analysis. Thus, the variation in DMSO content reflects assay design rather than a lack of standardization. The observation that DMSO concentrations up to approximately 6% (*v*/*v*) maintained enzymatic activity is consistent with previous studies on protein stability in mixed aqueous–DMSO systems. For instance, lysozyme unfolding studies indicated that moderate DMSO fractions cause only local perturbations in tertiary contacts, whereas global folding remains intact up to much higher solvent contents [29]. Similarly, α-chymotrypsin retained catalytic activity in low to intermediate DMSO levels, though kinetic constants shifted and destabilization occurred beyond this range [30]. Furthermore, moderate DMSO concentrations were shown to protect luciferase from proteolytic degradation [31], supporting its compatibility for enzyme-based assays. Collectively, these reports, along with the present findings, suggest that the BLuc–Red system remains structurally and functionally stable in the presence of low DMSO fractions, while higher levels lead to progressive inhibition due to solvent-induced destabilization of hydrophobic interactions.

While classical extraction solvents such as hexane or chloroform exhibit high efficiency toward non-polar hydrocarbons, they are inherently incompatible with the BLuc–Red enzymatic system due to their cytotoxicity and protein-denaturing effects [9]. Therefore, direct benchmarking of extraction efficiency was beyond the scope of this study, as conventional solvents cannot be introduced into the enzymatic assay without compromising enzymatic activity. Instead, the present work focused on evaluating DMSO as a biocompatible extractant that enables coupling of hydrocarbon solubilization with functional toxicity measurements. Future studies may incorporate indirect comparison approaches, including solvent exchange techniques or parallel chemical quantification, to quantitatively benchmark extraction yields against standard methods.

The inhibitory effect of diesel hydrocarbons dissolved in DMSO further emphasizes the sensitivity of the enzymatic bioluminescent assay. Even very low concentrations of hydrocarbons markedly reduced luminescence, reflecting the inherent toxicity of diesel constituents. Similar responses of bacterial luciferase to petroleum hydrocarbons and related aromatic compounds have been previously reported [32,33,34]. The clear dose–response relationship observed in DMSO soil extracts demonstrates that this solvent enables quantitative assessment of soil toxicity, where traditional aqueous bioassays frequently underestimate the impact due to poor hydrocarbon solubility.

To further interpret the inhibitory effects observed in the enzymatic assays at DMSO concentrations greater than 6% (*v*/*v*), MD simulations were performed to provide mechanistic insights into the molecular-level changes that may occur. Global stability parameters — RMSD, Rg, and SASA —remained stable across solvent conditions, indicating that neither DMSO nor diesel induced large-scale unfolding of luciferase or reductase. Slight increase in SASA for luciferase in the diesel-containing system indicated that more residues became exposed to solvent, although no significant loss of compactness was detected. RMSF analysis revealed localized rigidification within the α-subunit of luciferase (residues 145–155 and 281–287), corresponding to loops involved in enzyme dimerization and substrate fixation in the flavin-binding cavity [35]. For NAD(P)H:FMN-oxidoreductase, moderate flexibility changes were observed within residues 90–134 of subunit A and in the C-terminal loop of subunit B, which participates in phosphate fixation of the FMN cofactor [36,37]. Hydrogen-bond analysis showed a minor reduction in protein–water hydrogen bonds upon DMSO addition, both for the entire protein and for active-site residues, suggesting that DMSO competes with water molecules for hydration sites. These changes were more pronounced under diesel-containing conditions, likely reflecting partial rearrangement of hydrophobic contacts. Similar stabilizing effects at low DMSO fractions and disruption at higher levels have been reported for other enzyme systems [38,39,40].

Hence, local rigidification of luciferase loops and partial dehydration of both surface and active-site residues are consistent with the 1.6-fold inhibition observed at 9% DMSO. Although global metrics such as Rg and SASA do not indicate any large-scale structural rearrangements, local changes in BLuc loop mobility, as well as reduced water occupancy in the active center—which mediates intermediate transformations [41,42] – are critical for the bioluminescent reaction and may impair catalytic efficiency. Taken together, these observations support the conclusion that DMSO modulates luciferase activity primarily through local dynamical and hydration effects rather than through global structural changes.

From an ecotoxicological perspective, these results underscore the methodological value of DMSO for hydrocarbon toxicity testing. Conventional extraction approaches use solvents such as hexane or carbon tetrachloride, which, while effective at dissolving hydrocarbons, are highly toxic and incompatible with biological systems [9]. Conversely, aqueous bioassays tend to underestimate toxicity due to limited solubility of nonpolar compounds. The present work shows that DMSO offers a safe and effective compromise, ensuring both extraction efficiency and enzymatic compatibility. This methodological advance can enhance the comparability between chemical and bioassay-based toxicity evaluations, particularly relevant for oil-producing regions such as the Khanty-Mansi Autonomous Okrug-Yugra, where chronic hydrocarbon pollution and cold climates slow natural attenuation.

Certain limitations of this study should be acknowledged. Diesel fuel served as a model pollutant; however, crude oil, polycyclic aromatic hydrocarbons (PAHs), and aged hydrocarbon residues may differ in solubility and toxicity profiles. Soil heterogeneity, organic matter content, and co-contaminants such as metals can influence both extraction efficiency and bioluminescent responses. Moreover, while laboratory spiking ensures controlled conditions, natural soils experience weathering and microbial transformations that alter contaminant composition and toxicity over time [43,44,45]. The use of freshly spiked soils allowed strict control over contamination levels and soil matrix uniformity, which is essential for mechanistic evaluation of solvent–enzyme interactions. However, we acknowledge that naturally contaminated soils often contain weathered hydrocarbons, aged sorption complexes, and secondary metabolites that may modify extractability and toxicity. Incorporating field-contaminated soils will be an important next step to increase ecological relevance and to assess the applicability of the DMSO-based protocol to complex, long-term pollution scenarios. Future work should extend validation to field-contaminated soils, optimize DMSO concentrations for different contamination types, and explore the applicability of the method for complex and mixed pollutants. Previous studies have shown that matrix-induced inhibition of bioluminescence is highly soil-type dependent. In particular, pronounced matrix-related suppression of the BLuc–Red signal has been observed mainly in soils with high humus content, where humic substances can accelerate NADH consumption and attenuate luminescence independently of pollutant levels [14]. In contrast, low-humus sandy and sandy-loam soils demonstrate minimal intrinsic interference, and the luminescence response primarily reflects the concentration of hydrophobic toxicants. The Albic Podzolic soil used in the present study contains only 0.5% organic matter and exhibits weak humification; therefore, the contribution of humus-related matrix effects is expected to be negligible. This substantially reduces the likelihood of false-positive inhibition and supports the reliability of the DMSO-based extracts for assessing hydrocarbon-derived toxicity in this soil type.

In summary, this study provides both experimental and computational evidence supporting the use of DMSO as a biocompatible solvent for enzymatic bioluminescent assays of diesel-contaminated soils. By combining biochemical testing with MD simulations, we confirmed that moderate DMSO concentrations efficiently extract hydrocarbons while preserving enzyme structure and catalytic activity. This integrative approach strengthens the methodological basis of enzymatic bioluminescent assays and advances their application as sensitive tools for ecotoxicological risk assessment in petroleum-impacted environments.

## 5. Conclusions

This study demonstrated that DMSO can be effectively applied as a biocompatible solvent for enzymatic bioluminescent assays aimed at evaluating the toxicity of diesel-contaminated soils. Moderate DMSO concentrations preserved the catalytic activity of the coupled NAD(P)H:FMN-oxidoreductase and bacterial luciferase system while enabling efficient solubilization of hydrophobic diesel hydrocarbons. Molecular dynamics simulations confirmed that DMSO and diesel did not cause significant conformational destabilization of the enzymes, supporting their structural integrity under assay conditions. The combined experimental and computational findings validate DMSO as an extractant that bridges chemical extraction and biological assessment, thus improving the reliability of hydrocarbon toxicity evaluation in soil environments. This approach provides a methodological basis for integrating enzymatic bioassays into ecotoxicological monitoring and risk assessment frameworks, particularly for petroleum-impacted regions in cold climates. Future research should expand validation to field-contaminated and mixed-pollution soils and optimize solvent concentrations for routine environmental testing.

## Figures and Tables

**Figure 1 toxics-13-01038-f001:**
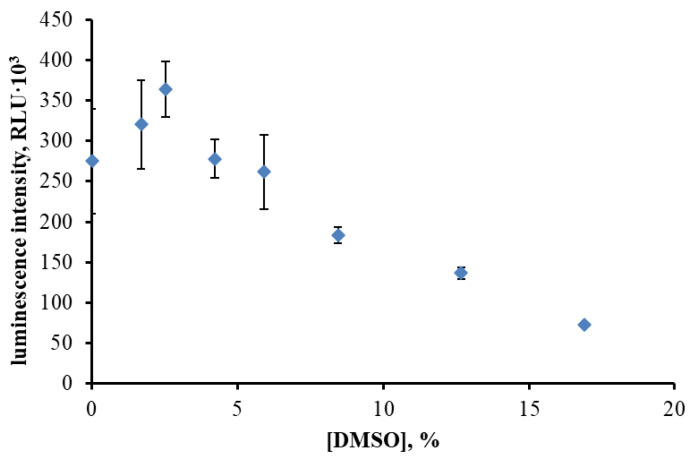
Effect of DMSO concentration on the maximum luminescence intensity of the BLuc–Red system.

**Figure 2 toxics-13-01038-f002:**
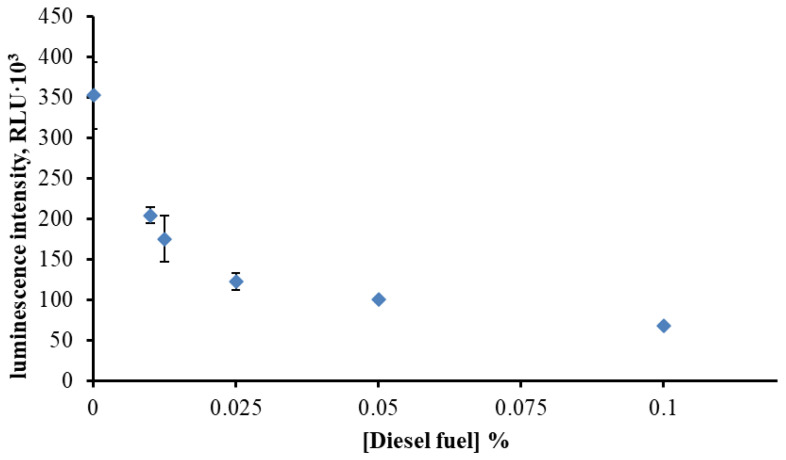
Effect of diesel fuel concentration (dissolved in 10% DMSO) on the maximum luminescence intensity of the BLuc–Red system.

**Figure 3 toxics-13-01038-f003:**
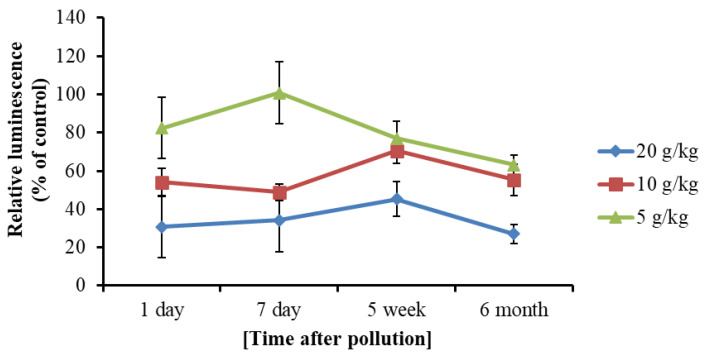
Residual activity of the BLuc–Red system in the presence of soil extracts obtained from diesel-contaminated samples.

**Figure 4 toxics-13-01038-f004:**
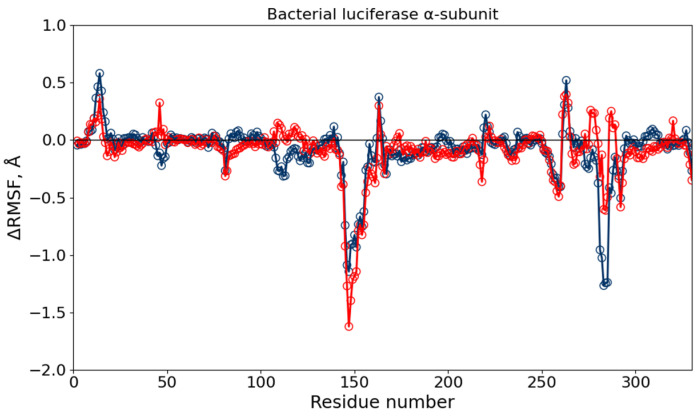
ΔRMSF of Cα atoms of the BLuc α-subunit in different solutions, calculated relative to the system in water. Blue empty markers show the difference between RMSF in 10% DMSO aqueous solution and water, red empty markers—between 10% DMSO aqueous solution with diesel and water. The negative ΔRMSF corresponds to a more rigid segment as compared with the structure in water.

**Figure 5 toxics-13-01038-f005:**
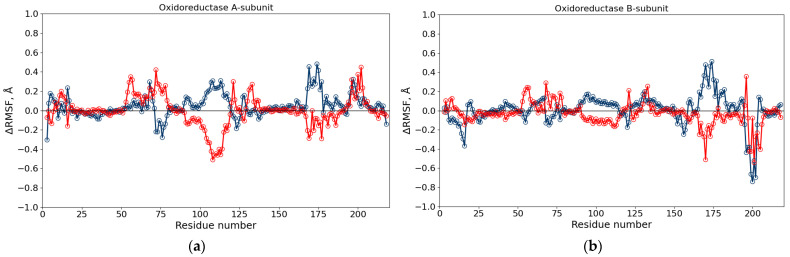
ΔRMSF of Cα atoms of the Red subunit A (**a**) and subunit B (**b**) in different solutions, calculated relative to the system in water. Blue empty markers show the difference between RMSF in 10% DMSO aqueous solution and water, red empty markers—between RMSF in 10% DMSO aqueous solution with diesel and water. The negative ΔRMSF corresponds to a more rigid segment as compared with the structure in water.

**Figure 6 toxics-13-01038-f006:**
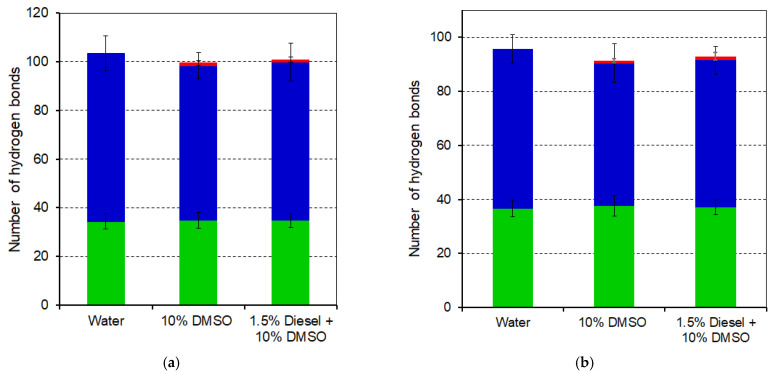
Number of hydrogen bonds formed by active site residues with other protein residues (green), water (blue), and DMSO (red) in the three simulated systems for BLuc (**a**) and Red subunit A (**b**).

**Table 1 toxics-13-01038-t001:** RMSD, SASA, and Rg values for luciferase and reductase in water, 10% DMSO, and 10% DMSO + 1.5% diesel fuel.

Parameter	Protein	Water	10% DMSO	10% DMSO + 1.5% Diesel
RMSD, Å	Luciferase	2.2 ± 0.2	2.1 ± 0.1	2.1 ± 0.1
	Reductase	2.1 ± 0.5	1.7 ± 0.3	1.9 ± 0.4
Rg, Å	Luciferase	27.0 ± 0.1	27.0 ± 0.1	27.0 ± 0.1
	Reductase	21.9 ± 0.1	21.8 ± 0.1	21.9 ± 0.1
SASA × 10^2^, Å^2^	Luciferase	297 ± 6	297 ± 4	302 ± 5
	Reductase	210 ± 4	208 ± 3	210 ± 3

## Data Availability

Data are contained within the article.

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
