# Peer review of "Dimethyl Sulfoxide as a Biocompatible Extractant for Enzymatic Bioluminescent Toxicity Assays: Experimental Validation and Molecular Dynamics Insights"

_toxics, 2025, doi:10.3390/toxics13121038_

Round 1
Reviewer 1 Report
Comments and Suggestions for Authors
Some clarifications and suggestions are needed to further improve the manuscript.
1) It is not easy to evaluate its true analytical advantage without benchmarking the extraction efficiency vs standard methods.
2) Validation using naturally contaminated soils would increase ecological relevance rather than using spiked soil samples.
3) Different DMSO concentrations of 4-6, 10, and 25% were used, raising questions about standardization, optimization, and potential solvent-enzyme interactions at higher percentages.
4) While simulations show structural stability, they are not directly correlated to measured decreases in luminescence, leaving some gaps between molecular insights and functional outcomes.
5) The assay assumes hydrocarbon-driven inhibition, but soil matrices contain many potential interferences that may affect luminescence. Indeed, it leads to false positives or overestimation of toxicity.
Author Response
REVIEWER #1 COMMENTS
Comment #1: It is not easy to evaluate its true analytical advantage without benchmarking the extraction efficiency vs standard methods.
Answer #1: We thank the reviewer for this important comment. The purpose of this study was to evaluate the biocompatibility and functional applicability of DMSO in enzymatic bioluminescent assays rather than to compare its extraction efficiency with that of classical organic solvents. As noted, conventional solvents commonly used for hydrocarbon extraction (hexane, chloroform, CCl₄) cannot be introduced into the BLuc–Red enzymatic system due to their strong protein-denaturing and cytotoxic effects. Therefore, direct benchmarking would not be methodologically valid.
To address the reviewer’s concern, we clarified this limitation and expanded the Discussion to explain why conventional extraction cannot be used in enzymatic assays and how future studies may compare extraction efficiency indirectly (e.g., solvent exchange or parallel chemical analysis) (lines 353-362).
Comment #2: Validation using naturally contaminated soils would increase ecological relevance rather than using spiked soil samples.
Answer #2: We agree with the reviewer that field-contaminated soils often contain weathered hydrocarbons, aged sorption complexes, and secondary transformation products that influence extractability and toxicity. In this study, freshly spiked soils were intentionally used to ensure strict control over contamination levels, soil uniformity, and solvent-enzyme interaction analysis.
We have added statements acknowledging this limitation and highlighted that assessment of naturally contaminated soils will be a priority in future research (lines 417-424).
Comment #3: Different DMSO concentrations of 4-6, 10, and 25% were used, raising questions about standardization, optimization, and potential solvent-enzyme interactions at higher percentages.
Answer #3: Thank you for raising this point. The different DMSO concentrations were intentionally applied for different methodological purposes:
1) 4-6%: assessment of BLuc-Red intrinsic solvent tolerance;
2) 10%: dissolution of diesel fuel for model hydrocarbon exposure tests;
3) 25%: extraction of soil samples to ensure effective solubilization of hydrophobic contaminants.
We clarified this rationale in the revised manuscript to avoid any misunderstanding and ensure transparency in assay design (lines 333-340).
Comment #4: While simulations show structural stability, they are not directly correlated to measured decreases in luminescence, leaving some gaps between molecular insights and functional outcomes.
Answer #4: We appreciate this constructive observation. We expanded the Discussion to clarify that molecular dynamics simulations were used to uncover mechanistic underpinnings rather than to quantitatively predict catalytic rates. Specifically, the simulations revealed localized rigidification and partial dehydration of key luciferase regions, which is consistent with the observed moderate inhibition of luminescence at higher DMSO concentrations.
This explicitly links molecular-level changes with functional assay outcomes (lines 371-399).
Comment #5: The assay assumes hydrocarbon-driven inhibition, but soil matrices contain many potential interferences that may affect luminescence. Indeed, it leads to false positives or overestimation of toxicity.
Answer #5: We fully agree that soil matrices can contain humic substances, metals, salts, and other components capable of interfering with luminescence-based assays. To address this point, we added a dedicated Discussion paragraph that (i) recognizes the potential for matrix-induced effects, (ii) cites earlier work showing how humus-rich soils may cause signal suppression, and (iii) explains why such interference is minimal in the present study (lines 426-472).
Importantly, our soil type (Albic Podzolic) contains only 0.5% organic matter and exhibits weak humification. Based on previous research, matrix interference primarily affects strongly humified soils, while low-humus sandy or sandy-loam soils show minimal intrinsic suppression of the BLuc–Red signal. We have therefore clarified why false-positive inhibition is unlikely for the specific soil type used in our experiments.
Reviewer 2 Report
Comments and Suggestions for Authors
The manuscript presents a valuable investigation into the use of Dimethyl Sulfoxide (DMSO) as a compatible extractant for enzymatic bioluminescent assays applied to diesel-contaminated soils. The integration of experimental bioassays with molecular dynamics (MD) simulations is a commendable approach that provides a multi-faceted perspective on solvent-enzyme interactions. The topic is highly relevant to the field of ecotoxicology, particularly for monitoring petroleum-impacted sites. However, several major concerns regarding the experimental design and the interpretation of results relative to the core hypothesis must be addressed before the manuscript can be considered for publication. I therefore recommend Major Revision.
- It is unclear whether soil extracts were diluted before the bioassay. If so, please specify the dilution factor and its impact on the final DMSO concentration in the assay mixture.
- The diesel composition was adopted from a reference, but it would be helpful to briefly describe the main components used in the simulation (e.g., alkanes, aromatics) to aid interpretation.
- The MD simulations conclude that neither 10% DMSO nor diesel induces significant conformational changes in the enzymes, with only localized flexibility alterations reported. This stands in direct contrast to the experimental data, which show a ~1.6-fold reduction in luminescence intensity in 9% DMSO. The manuscript fails to provide a mechanistic explanation for this apparent contradiction. How can the enzymatic activity be significantly compromised without corresponding large-scale or functionally critical structural perturbations?
- The use of freshly spiked diesel soil in a laboratory setting does not fully represent the complexity of aged, field-contaminated soils.
- Figure 3: The Y-axis is labelled "Residual activity (%)", yet the values exceed 100%. Please verify the data and correct the axis label or values accordingly.
- Table 1: "Lusiferase" is a typo and should be corrected to "Luciferase".
- MD Methods: While the diesel composition is referenced, a brief description of the major hydrocarbon classes (e.g., alkanes, aromatics) used in the model would enhance reproducibility and transparency.
Author Response
REVIEWER #2 COMMENTS
Comment #1: It is unclear whether soil extracts were diluted before the bioassay. If so, please specify the dilution factor and its impact on the final DMSO concentration in the assay mixture.
Answer #1: We thank the reviewer for this important clarification request. The soil extracts prepared using 25% (v/v) DMSO were introduced directly into the reaction mixture without additional dilution. Given that 150 µL of extract was added to a total reaction volume of 420 µL, the final DMSO concentration in the enzymatic assay was approximately 9% (v/v). This information has now been explicitly added to the Materials and Methods section (lines 187-190).
Comment #2: The diesel composition was adopted from a reference, but it would be helpful to briefly describe the main components used in the simulation (e.g., alkanes, aromatics) to aid interpretation.
Answer #2: We appreciate the reviewer’s suggestion. A concise description of the representative hydrocarbon classes included in the diesel model has been added for clarity and reproducibility. These include n-paraffins, isoparaffins, naphthenes, aromatics, and heteroaromatics (lines 208-211).
Comment #3: The MD simulations conclude that neither 10% DMSO nor diesel induces significant conformational changes in the enzymes, with only localized flexibility alterations reported. This stands in direct contrast to the experimental data, which show a ~1.6-fold reduction in luminescence intensity in 9% DMSO. The manuscript fails to provide a mechanistic explanation for this apparent contradiction. How can the enzymatic activity be significantly compromised without corresponding large-scale or functionally critical structural perturbations?
Answer #3: We thank the reviewer for raising this important question. We fully agree that the observed ~1.6-fold decrease in luminescence at 9% DMSO reflects a functionally relevant perturbation. Importantly, efficient catalysis in bacterial luciferase depends not on large-scale structural rearrangements, but on finely tuned local dynamics within the active-site loop and on the maintenance of a specific hydration network in the catalytic pocket.
While our MD simulations did not reveal global unfolding or pronounced conformational distortions, they did detect localized yet catalytically meaningful changes. These include (i) reduced mobility of the active-site loop and (ii) partial dehydration of key residues within the FMN-binding cavity. Both effects are well documented in the literature as critical modulators of luciferase turnover because loop flexibility controls access and stabilization of reaction intermediates, while structured water molecules participate directly in the oxidative transformation steps.
Thus, the absence of large-scale structural changes is not contradictory to the experimental inhibition; instead, the MD results reveal precisely the type of subtle dynamical and hydration perturbations that are known to reduce quantum yield and reaction rate without compromising global protein stability.
To further support this mechanistic link, we have strengthened the Results by adding quantitative analysis of hydrogen-bonding patterns between active-site residues and solvent molecules (Lines 309–315). We also expanded the Discussion to explicitly connect these localized dynamical effects with the measured decrease in luminescence intensity (Lines 371–399).
Comment #4: The use of freshly spiked diesel soil in a laboratory setting does not fully represent the complexity of aged, field-contaminated soils.
Answer #4: We agree with the reviewer. This limitation is now more clearly acknowledged in the revised Discussion. Freshly spiked samples were used to maintain full control over contamination levels and to isolate solvent-enzyme interactions. We now explicitly state that weathered pollutants, aged sorption complexes, and mixed contaminants typical of field soils must be evaluated in future work (lines 417-424).
Comment #5: Figure 3: The Y-axis is labelled "Residual activity (%)", yet the values exceed 100%. Please verify the data and correct the axis label or values accordingly.
Answer #5: We thank the reviewer for pointing this out. The slight exceeding of 100% resulted from the upper bounds of standard deviation, not from the measured mean values. To avoid confusion, we have corrected the Y-axis label to “Relative luminescence (% of control)”, which accurately reflects the normalization method (I/I₀ × 100) and naturally allows statistical error to extend above 100%.
Comment #6: Table 1: "Lusiferase" is a typo and should be corrected to "Luciferase".
Answer #6: Corrected. Thank you for noting this typographical error.
Comment #7: MD Methods: While the diesel composition is referenced, a brief description of the major hydrocarbon classes (e.g., alkanes, aromatics) used in the model would enhance reproducibility and transparency.
Answer #7: We appreciate this suggestion. As noted in answer #2, we added a clear description of the included hydrocarbon classes to ensure full transparency.
Round 2
Reviewer 2 Report
Comments and Suggestions for Authors
The author carefully responded to the review comments and suggested accepting the manuscript.